# Workforce predictive risk modelling: development of a model to identify general practices at risk of a supply–demand imbalance

Gary A Abel  ,[1] Mayam Gomez-Cano,[1] Navonil Mustafee,[2] Andi Smart,[2] Emily Fletcher  ,[1] Chris Salisbury  ,[3] Rupa Chilvers,[4] Sarah Gerard Dean  ,[5] Suzanne H Richards,[6] F Warren,[1] John L Campbell[1]

[1]University of Exeter Medical School (Primary Care), University of Exeter, Exeter, UK
[2]University of Exeter Business School, Exeter, UK
[3]Centre for Academic Primary Care, NIHR School for Primary Care Research, School of Socialand Community Medicine, University of Bristol, Bristol, UK
[4]Tangerine Bee, Exeter, UK
[5]PenCLAHRC University of Exeter Medical School, University of Exeter, Exeter, UK
[6]Academic Unit of Primary Care, University of Leeds, Leeds, UK

**Correspondence to**
Dr Gary A Abel;
g.a.abel@exeter.ac.uk

## ABSTRACT

**Objective** This study aimed to develop a risk prediction model identifying general practices at risk of workforce supply–demand imbalance.

**Design** This is a secondary analysis of routine data on general practice workforce, patient experience and registered populations (2012 to 2016), combined with a census of general practitioners' (GPs') career intentions (2016).

**Setting/Participants** A hybrid approach was used to develop a model to predict workforce supply–demand imbalance based on practice factors using historical data (2012–2016) on all general practices in England (with over 1000 registered patients n=6398). The model was applied to current data (2016) to explore future risk for practices in South West England (n=368).

**Primary outcome measure** The primary outcome was a practice being in a state of workforce supply–demand imbalance operationally defined as being in the lowest third nationally of access scores according to the General Practice Patient Survey and the highest third nationally according to list size per full-time equivalent GP (weighted to the demographic distribution of registered patients and adjusted for deprivation).

**Results** Based on historical data, the predictive model had fair to good discriminatory ability to predict which practices faced supply–demand imbalance (area under receiver operating characteristic curve=0.755). Predictions using current data suggested that, on average, practices at highest risk of future supply–demand imbalance are currently characterised by having larger patient lists, employing more nurses, serving more deprived and younger populations, and having considerably worse patient experience ratings when compared with other practices. Incorporating findings from a survey of GP's career intentions made little difference to predictions of future supply–demand risk status when compared with expected future workforce projections based only on routinely available data on GPs' gender and age.

**Conclusions** It is possible to make reasonable predictions of an individual general practice's future risk of undersupply of GP workforce with respect to its patient population. However, the predictions are inherently limited by the data available.

### Strengths and limitations of this study

► This study made use of freely available data from a range of sources to develop a predictive model of workforce supply–demand imbalance for general practices in England.
► Historical data for all of England is used to develop factor weightings which are then applied to current data.
► The additional value of a census survey of career intentions of general practitioners in South West England is explored, comparing findings to predictions made on the basis of general practice workforce age and gender alone.
► The predictive model is inherently limited by the data available, and in particular we note that routine data of a measure of a practice's difficulty in recruiting staff were not available.

## INTRODUCTION

Against a backdrop of 34 495 full-time equivalent (FTE) general practitioners (GPs) in 2016, the National Health Service (NHS) in England saw a reduction of 3.5% of the English GP workforce (1193 FTE) in a single year.[1] This reduction has been seen in combination with rising demands of the patient population.[2] Such figures represent a 'crisis' in respect of GP workforce capacity, with particular problems in retaining established GPs in direct patient care.[3 4] Similar problems in respect of family doctor recruitment and retention are evident in other western healthcare economies and jurisdictions,[5 6] and many countries have explored what might constitute optimal skill mix among primary care health professionals over the last 40 years.[7–9]

There is, however, a need for the rational deployment of the GP workforce resource.[10 11] Various models exist to inform that deployment, with GP workload representing a key

issue among individual GPs electing to quit patient care.[3] Gaining an understanding of GP workload pressures is also the basis of identifying any potential mismatch between the demand for general practice services and the supply of GPs to meet that demand. In many countries, the general practice represents a key element in the delivery of primary care and acts as the basis for general practice workforce planning. For example, practices are the basis of reporting of patients' experience of primary care in England, captured using the General Practice Patient Survey (GPPS).[12]

The aim of this research was to develop a method to identify NHS general practices in one region of England which may face supply–demand workforce imbalances within the next 5 years. Previous workforce modelling in the UK has focused on deriving insights from analyses at the regional or national (macro) level.[13] In contrast, the research we are reporting here focuses on undertaking predictive risk modelling at a practice (micro) level. Routine workforce modelling makes use of data on doctors' age and gender, and historical retirement patterns. Here, we consider whether surveying GPs' career intentions adds value to such modelling.

The first step in developing a predictive model to identify general practices at risk of future supply–demand imbalance is to define what is meant by a supply–demand imbalance and to operationalise this with measurable quantities. Assessing the supply of GP workforce at any one general practice is reasonably straightforward; however, assessing the demand of patients is complex as unmet demand is, by its nature, hard to quantify. Instead, here we consider the expected workload given the demographics of the patient population served. The balance between supply and demand within this framework is then represented by the expected workload per practitioner. However, high workload alone may not be an issue. Practices with high workload may meet patient demand through innovative and efficient systems of service delivery. High workload is considered to have a negative impact only when service delivery is impaired. For the purposes of this study, we defined those practices with high workload per practitioner in combination with an inability to meet patient demand as being in a state of 'undersupply'. Here, we use the term 'undersupply' to indicate a practice which has a high demand from patients for a given supply of doctors which appears to be having a detrimental impact on services.[14] In this study, we used a measure of patient access as a proxy for the ability to meet patient demand, in the belief that access is an important measure, reflecting the ease with which patients might engage with the primary healthcare system.[14 15]

## METHODS
### Overview
Several data sources have been brought together in this work. Analyses were performed at general practice level, first, to identify practices which were currently in 'undersupply' and, second, to identify those which are likely to have such problems in the future. A predictive risk model (to predict the risk of a practice being in a state of 'undersupply' within 5 years) was developed by assessing the associations between current (2016) 'undersupply' status and historical routinely collected data (where available) on GP workforce, practice characteristics (rurality, deprivation, population) and patient experience scores from 2012. The model further incorporated projected future populations in each area and considered projected future GP workforce based on GPs stated career intentions (from a survey of GPs). The rationale for this approach was to obtain factor weightings informed by evidence developed on past data. This model was then used to identify practices and areas in South West England that are likely to experience a supply–demand imbalance ('undersupply') in the future.

### Data sources
Except where specified, national data for England were obtained and processed. A summary of data sources is given in the following paragraphs with full details given in online supplementary appendix 1, along with a schematic illustrating the data flow used in the modelling process (online supplementary appendix 2).

### General Practice Patient Survey (GPPS)
The GPPS is a national postal survey of patients' experience of primary care in England distributed to around 2.8 million adult patients each year.[12] We used data from the 2011/2012 and 2015/2016 surveys, during which the contents of the survey remained largely consistent. Response rates were 38% in 2011/2012 (1 037 946 responses) and 39% in 2015/2016 (836 312 responses) with an average of around 125 respondents per practice.

### Workforce
Workforce data at practice level were obtained from National Health Service (NHS) Digital and related to GP Census data taken as at 30 September 2012, 2013 and 2016.[16–18]

### General practitioner quitting intentions
Self-reported GP intentions to cease practice were collected through a census survey which has been reported elsewhere.[19] Briefly, a questionnaire was administered to all active GPs in South West England in April–June 2016, enquiring about their intentions to cease/interrupt practice within 2 and 5 years (3370 questionnaires sent, 2248 returned, response rate 67%).

### Practice rurality and deprivation
Practice rurality (rural/urban) based on an Office for National Statistics (ONS) categorisation of the postcode of the practice was obtained, as was a practice deprivation score based on the 2015 Index of Multiple Deprivation (IMD).[20]

## Practice registered population

Data on the registered populations for each general practice were obtained for each quarter from April 2014 to April 2016 (nine datasets), as well as April 2012. These datasets provided the count of patients of each gender (male, female) by 5-year age-band strata.

## Subnational population projections

We made use of the ONS subnational population projections developed to inform the local planning of healthcare and other public services for geographically defined populations served by Clinical Commissioning Groups (CCGs, organisations responsible for commissioning NHS services).[21] These projections are demographic, trend-based projections that indicate the 'likely levels of future population' and are routinely produced every 2 years. We extracted projected populations for 2021 for the eight CCGs within South West England. Projections were made in 5-year age-bands for each gender.

## Variables

Brief details are given in the following paragraphs with full details in online supplementary appendix 1

## Patient experience

We used three GPPS items reflecting access ("Last time you wanted to see or speak to a GP or nurse from your GP surgery: Were you able to get an appointment to see or speak to someone?"), continuity of care ("How often do you see or speak to the GP you prefer?") and overall experience ("Overall, how would you describe your experience of your GP surgery?"). Case-mix adjusted practice scores for patient experience were created following previous methodology[22][23] adjusting for patient age, gender, ethnicity, presence of a long-term condition and deprivation, using mixed effects logistic regression. The case-mix adjusted scores were based on dichotomous outcomes and used in the form of log ORs relative to the average practice nationally.

## Workforce

Practices with less than 0.5 GP FTE (38 out of 7484 practices in 2012 data and 41 out of 6709 practices in 2016 data) were excluded from all analyses on the basis that such a low staff record indicated either that these were unusual practices or that the workforce data were in error. In addition to total GP FTE, the ratio of nurse FTE to doctor FTE and the ratio of doctor FTE in the 'other' category to total doctor FTE were calculated (where 'other' is assumed to mostly be locum GPs given that registrars, salaried GPs and those on retainer schemes, are captured in specific categories). Total nurse FTE data were not available in 2012, so 2013 data were used in its place.

## Workload

We used a definition of workload based on registered patients rather than on recorded patient visits. Patient visits are a measure of actual work undertaken which is limited by the workforce available, and so cannot capture unmet demand. By focusing on the registered population, we estimated the expected workload to serve that population based on national averages. Weights were applied to patient list sizes in order to standardise for the age and gender composition of the practice population, accounting for the fact that GPs spend longer, on average, consulting with patients who are very young, are older or are female.[2] Further adjustment was made for the deprivation of the practice population to reflect higher health needs. These adjusted weighted list sizes were divided by the total GP FTE to obtain a measure of workload per GP FTE. Initial inspection of the workload figures showed that the distribution contained some infeasibly large and small values. Practices in the top and bottom 2.5% of the distribution were excluded from all further analysis. This exclusion took place following the removal of practices with less than 0.5 GP FTE.

## Expected remaining future workforce

We estimated the proportion of GP FTE that would be expected, on average, to remain in patient care in 5 years' time. We did this in two principal ways: (1) using information on the age and gender of GPs at the practice along with previous work which identified the probability that GPs of different ages and genders leave patient care[24] and (2) based on responses to survey of GP's career intentions. The former was done for both the 2012 and 2016 data and the latter only for the 2016 data. The approaches are detailed in full in online supplementary appendix 1.

## Outcome definition

Ability to meet patient demand was quantified using the GPPS access measure (ability to make an appointment), reflecting the ease with which patients might engage with the primary healthcare system. Workload to workforce ratio was quantified using the workload per GP FTE quantity described above. Practices that were in the lowest third of GPPS access scores and also in the highest third of workload per GP FTE nationally were defined as being in 'undersupply' (ie, demand exceeded supply). Having used relative measures and cut points which were defined pragmatically for the purposes of this study in our definition of undersupply, we do not propose absolute and objective measures about whether a practice is 'failing' to deliver care. Indeed, if provision of care were good everywhere and the supply of workforce were not an issue, such an approach would be inappropriate. However, in the current climate in the UK, this represents a pragmatic approach in the absence of a direct measure.

## Development of predictive risk model

Historical data were used to produce model coefficients which could then be applied to current data. Model development was based on all available national data in order to maximise statistical power. We did not split the data into development and validation samples as changes over time in healthcare delivery are more likely to be a threat to future use of the model than overfitting. Predictor

variables (as shown in online supplementary appendix 2a) were based on the 2012 data unless otherwise noted and included:

► Three GPPS scores.
► Adjusted weighted list size per GP FTE (workforce to workload ratio).
► Total GP FTE.
► The ratio of 'other' GP FTE to total GP FTE.
► The expected proportion of GP FTE still in patient care in 2017.
► Ratio of nurse FTE to doctor FTE (using nurse FTE data from 2013).
► 2016 adjusted weighted list size (using 2016 data).
► Rurality setting (based on the 2016 data, but not expected to change).
► Practice deprivation (based on the 2016 data, but not expected to change).

We did not attempt to predict the 2016 practice populations using only data available in 2012 and instead included the observed 2016 practice populations as an additional explanatory variable due to a lack of data available for 3 years prior to 2012.

A logistic regression model was used with a binary outcome of a practice being in a state of undersupply in 2016 based on the 2016 data (see outcome definition above). Practices were the unit of analysis. All variables considered were included and retained regardless of statistical significance.

We recognised the need to account for the fact that GPs leaving patient care would be most likely to impact the supply–demand balance when recruitment of staff was difficult. We were unable to obtain any direct measure of the difficulty any one practice had in recruitment and so instead we explored the use of three proxy measures:

1. The use of locums (operationalised as the proportion of total GP FTE falling in the 'Other' category using NHS workforce data), on the basis that practices are likely to make greater use of locums when they are struggling to recruit partners or salaried GPs.
2. Patient access (using GPPS scores), on the basis that when there is a prolonged period where a practice is understaffed access may be compromised.
3. The use of nurses (operationalised as the ratio of total nurse FTE to total GP FTE using NHS workforce data), on the basis that practices that have difficulty in recruiting GPs may employ more nurses to take on aspects of patient care traditionally delivered by GPs, thus freeing up GP time.

In exploratory analysis, an interaction between the expected proportion of the GP workforce remaining in patient care after 5 years and each of the identified proxy measures (use of locums, access, use of nurses) individually were included in the predictive model in turn. There was no evidence that either locum use or access modified the effect, in the model, of the expected proportion of the GP workforce remaining in patient care. However, there was weak evidence that the use of nurses did modify the effect of the expected proportion of the GP workforce

remaining in patient care. This interaction was, therefore, retained in the final model. The predictive value of our model was assessed using an ROC (receiver operating characteristic) curve analysis of predicted probabilities for all practices in England based on the data used to build the model (ie, 2012 data and 2016 supply–demand imbalance classifications). So as to improve the generalisability of our findings and account for the fact that there will be a degree of overfitting in our model, we employ 10-fold cross-validation to estimate the area under the ROC curve.[25] These were compared with a simpler model developed using only two explanatory variables which were the 2012 data for factors defining the undersupply (GPPS access scores and adjusted weighted list size per FTE, noting that the outcome of the model, undersupply, was still based on 2016 data, online supplementary appendix 2c). Calibration was assessed by comparing the mean predicted probability from the main model and the percentage of practices in undersupply in 2016 for deciles of predicted probability. We also performed a sensitivity analysis to examine the impact of excluding the top and bottom 2.5% of practices in terms of workload per GP FTE. To do so, we re-ran the logistic regression after excluding only the top and bottom 1% of practices in terms of workload per GP FTE.

### Future risk prediction

The coefficients from the historical model were applied to the 2016 data to form our baseline risk predictions with a 5-year forward view for practices in South West England only (as shown in online supplementary appendix 2b). The reason for the restriction to those practices was that they were the only ones for which we had survey responses on future career intentions. It should be noted that although the original outcome definition was a relative one, the model treated them as absolute. In other words, predictions obtained from the model identify the risk of having a workload to workforce ratio in 2021 higher than two-thirds of practices did in 2016 and a GPPS access score in 2021 lower than two-thirds of practices did in 2016. In the context of a nationally worsening situation, this would allow for considerably more practices to be in a state of undersupply. Practices in the highest 25% of the predicted risk profile were flagged as 'high risk' of future undersupply of GP workforce, those in the lowest 25% were flagged as being 'low risk' and those in between were flagged as being at 'moderate risk'.

The usefulness of the career intention survey was examined by comparing the above prediction with an alternative prediction using the expected proportion of the GP workforce remaining in patient care in 5 years' time based only on the routinely available age and gender profile of GPs in the practice.

In addition to baseline predictions, we explored a number of 'stress testing' scenarios. These scenarios can be considered as stress tests of the model to identify practices that might be more (or less) vulnerable to particular challenges. First, we explored the effect of increased

difficulty in recruiting GPs, which we modelled as an increase in the coefficient for the expected proportion of GPs remaining in patient care (where an increased coefficient implies a greater impact of GP workforce leaving patient care). Second, we explored which practices might be at particular risk of a marked increase in local population. This was done by inflating the predicted adjusted weighted list size. The following scenarios were explored:

A. The coefficient for expected proportion of GPs remaining in patient care increased by 2 (equivalent to a 22% increase in the odds of being in supply–demand imbalance when 10% of GPs are expected to leave representing a modest increase in the difficulty of recruiting GPs).

B. The coefficient for expected proportion of GPs remaining in patient care increased by 4 (equivalent to a 49% increase in the odds of being in supply–demand imbalance when 10% of GPs are expected to leave representing a substantial increase in the difficulty of recruiting GPs).

C. The predicted adjusted weighted list size increased by 20%.

D. The predicted adjusted weighted list size increased by 40%.

E. A modest increase in difficulty recruiting GPs combined with a 20% increase in list size (A and C combined).

F. A substantial increase in difficulty recruiting GPs combined with a 40% increase in list size (B and D combined).

For each of these scenarios, practices were rated according to relative risk (ie, top 25% were labelled 'high relative risk' as above) and absolute risk. The relative risk cut-offs in the baseline scenario were used for absolute risk cut-offs in the other scenarios.

### Patient and public involvement

This study was part of a wider programme of work considering GP workforce issues which was served by a Patient and Public Involvement (PPI) group which provided input to the overall design and conduct of the research. Developing methods and results were shared at project management group meetings, which included PPI representatives who directly contributed to refining methods, and interpreting and contextualising the results.

Analyses were performed using Stata V.14 and V.16, and the 10-fold cross-validation was performed using the CVAUROC command.

## RESULTS
### Mapping the current situation

A total of 6398 practices in England had available data on all data items and had list sizes >1000; 371 of these were in South West England. The distribution of practices in England as a whole and South West England is shown in figure 1. Practices with GPPS access scores (ability to make an appointment—our proxy for ability to meet patient demand) in the highest scoring third nationally were over-represented in South West England, with 57% of practices in this region falling in that category. There was also an under-representation of South West practices nationally in respect of workload (only 22% of practices in the region were classified as in the third of practices nationally with the highest workload). As a result, the percentage of practices defined as currently being in undersupply was considerably lower in South West England (5.1%) than in England as a whole (13.5%).

There was no evidence that list size varied between those practices in undersupply and other practices in South West England (table 1). However, there was evidence that practices in undersupply had fewer FTE GPs. Together, these findings indicate that observed differences in workload are driven more by the supply of GP workforce than the demand of the registered patient population. Practices in undersupply also had a higher ratio of nurse FTE to GP FTE, served more deprived populations, had lower patient experience scores, had fewer patients over the age of 65 and were more likely to be in urban areas.

### Predictive risk model

The regression coefficients for the logistic model are shown in table 2. Predictive risk model coefficients were estimated using the 2012 data where possible to estimate the independent association with 2016 undersupply status. A negative coefficient implies a reduced risk of future undersupply as the value of the variable increases when all other variables are kept constant. We note the interaction between the expected proportion of GP FTE still working in patient care in 5 years' time and the ratio of nurse FTE to doctor FTE had a relatively large p value (0.177). In initial modelling (before excluding practices on the basis of data quality), this interaction variable had a smaller p value (0.06), indicating some evidence that it was worth including. When exclusions were applied, the coefficient did not change meaningfully. This fact, combined with the a priori expectation that the effect of expected future GP workforce would be dependent on recruitment, provided support to retain the interaction term. The sensitivity analysis excluding only the top and bottom 1% of practices in terms of workload per GP FTE produced broadly similar regression coefficients with the exception of the coefficient for the expected proportion of GP workforce to remain in patient care which was reduced by 43% (results not shown).

Figure 2 shows the 10-fold cross-validation ROC curve derived from the development model (ie, 2012 covariates and 2016 outcome). The mean area under the curve was 0.755. The ROC curve from the simpler model only including the defining factors (GPPS access scores and adjusted weighted list size per FTE) had a mean area under the curve of 0.695, suggesting that the additional variables included in our model provided a modest, but meaningful, improvement in predictive value. A visual inspection of a calibration plot for the full model suggests

**Figure 1** Distribution of practices in England and in South West England across categories according to workforce to workload ratio and General Practice Patient Survey access scores. FTE, full-time equivalent; GP, general practitioner.

that there is good calibration of the model (online supplementary appendix 3).

### Future risk predictions

Applying the risk prediction model to data from 2016, seeking to predict the risk of future supply–demand imbalance for individual practices in South West England, we obtained risk scores for 368 practices with available data remaining after applying exclusions. The median probability of future supply–demand imbalance across practices was 5.4% (IQR 2.8% to 10.0%). In total, 40 (10.9%) practices had a risk greater than 20%, and 12 (3.3%) had a risk greater than 50%. Table 3 shows the characteristics of those practices in South West England classified as high risk (top 25% of practices, corresponding to an absolute risk of 10% or greater) of being in a state of under-supply compared with other practices. In contrast to the current situation shown in table 2, there was no evidence (p=0.445) that the total GP FTE varies between high/other risk classification. There was evidence, however,

that all other descriptive factors varied between the two groups. Practices at 'high risk' of future supply–demand imbalance tended to currently have larger list sizes, to have a higher proportion of nurses in the workforce, to serve more deprived and younger populations, have considerably worse GPPS scores and were more likely to be in urban areas.

### Stress testing scenarios

Figures 3 and 4 illustrate the changes to the relative and absolute risk of future undersupply under different stress testing scenarios. In this figure, each practice is represented by a horizontal bar. The vertical ordering of each practice is the same in each scenario and is based on the rank ordering of each practice according to the baseline risk prediction. For each scenario, the colouring of every practice's horizontal bar illustrates the relative or absolute risk classification (figures 3 and 4, respectively) such that changes in colour indicate changes in risk classification. In figure 3, practices coloured red (high risk) are in

**Table 1** Comparison of practices in South West England defined as in undersupply with other practices in the region

| | Undersupply (n=19) | | | Other (n=352) | | | |
|---|---|---|---|---|---|---|---|
| | Median | 25% | 75% | Median | 25% | 75% | P value* |
| List size | 9264 | 5361 | 11576 | 7598 | 5270 | 11077 | 0.448 |
| Adjusted weighted list size | 8959 | 5212 | 12287 | 8099 | 5638 | 11570 | 0.550 |
| GP FTE | 3.1 | 2 | 5.1 | 4.7 | 3.2 | 6.6 | 0.012 |
| Ratio nurse/GP FTE | 0.8 | 0.7 | 1 | 0.5 | 0.4 | 0.7 | <0.001 |
| Index of Multiple Deprivation§ | 25.7 | 20.2 | 30.9 | 18.7 | 13.5 | 24.4 | 0.003 |
| GPPS access† | 0.2 | 0.1 | 0.2 | 0.7 | 0.5 | 0.9 | <0.001 |
| GPPS continuity† | 0.2 | 0.2 | 0.3 | 0.6 | 0.4 | 0.8 | <0.001 |
| GPPS satisfaction† | 0.2 | 0.1 | 0.4 | 0.7 | 0.5 | 0.9 | <0.001 |
| % over 65 | 16.8 | 13.3 | 21 | 22.6 | 17.6 | 26 | 0.004 |
| Setting | n | % | | n | % | | P value‡ |
| Urban practices | 17 | 6.8 | | 232 | 93.2 | | 0.042 |
| Rural practices | 2 | 1.6 | | 120 | 98.4 | | |

*From Mann-Whitney test.
†GPPS scores used were case-mix adjusted log ORs relative to the average practice nationally.
‡From Fisher's exact test.
§Index of Multiple Deprivation scores are given (rather than ranks) with higher scores indicating higher levels of deprivation.
FTE, full-time equivalent; GP, general practitioner; GPPS, General Practice Patient Survey.

the top 25% of practices in terms of risk of undersupply for any given scenario, practices coloured green (low risk) are in the bottom 25% for any given scenario, with the middle 50% of practices coloured yellow. In figure 4, practices coloured red (high risk) have an absolute risk of future undersupply greater than 10% (corresponding to the minimum absolute risk of future undersupply of the top 25% of practices in the baseline scenario), practices coloured green (low risk) have an absolute risk less than 2.8% (corresponding to the maximum absolute risk of the bottom 25% of practices in the baseline scenario) and intermediate practices are coloured yellow.

First, we examined the changes in predictions when using the two different methods of quantifying the likely future GP workforce remaining in patient care (one method using the results of the career intention survey and one method using only GP age and gender). The two methods produced similar values for the likely proportion of GP workforce remaining in patient care with a Spearman correlation of 0.77 between the estimates made using the two methods in the 387 practices with at least one survey response. When using the different methods in the risk prediction model, there was very little difference in practices categorised as being either at 'high relative risk' or at 'high absolute risk' of undersupply (seen in figure 4 as limited reclassification of practices, correlation of ranks=0.999).

In general, practices classified as being at 'high relative risk' remained so under scenario A (modest increase in the difficulty of GP recruitment to replace those leaving—correlation in ranks between scenario A and baseline=0.97). However, there was a dramatic increase in the number of practices with a predicted absolute

risk of future undersupply greater than 10% (seen as an increase in the number of practices coloured red figure 4, scenario A). There was an even greater disturbance in the classification of practices under scenario B (illustrating the recruitment of GPs was becoming much harder), although the reclassification in terms of relative risk was still relatively modest (figure 3, scenario B, correlation in ranks between scenario B and baseline=0.90). Conversely, the reclassification in terms of absolute risk (figure 4, scenario B) was significantly greater; the majority of practices had a predicted risk above 10%.

Increasing the projected practice population resulted in only modest changes in respect of which practices are classified as being at 'high relative risk'. Only a small relative increase was seen when comparing scenarios C and D with the baseline predictions (figure 3 correlation in ranks between scenario C and baseline=0.99 and scenario D and baseline=0.98). However, substantial changes were seen in the number of practices with an absolute risk of undersupply greater than 10% (figure 4, scenarios C and D). Combining the effect of scenarios A and C resulted in relative risk classifications closer to the baseline predictions than scenario A alone. However, in terms of absolute risk, more practices had a risk greater than 10% (figure 4, scenario A and scenario C).

When scenario B and scenario D were combined (illustrating a situation where it was much harder to recruit GPs combined with an increased practice population of 40%), it was evident that nearly all practices (88%) exceeded 10% absolute risk of supply–demand imbalance within 5 years, with only nine (2.4%) practices classified as being at 'low absolute risk' using the cut-offs derived from the baseline predictions.

**Table 2** Predictive risk model coefficients estimated using the 2012 data where possible to estimate the independent association with 2016 undersupply status

| Data type | Variable | Note on units | Logistic regression coefficient (95% CI) | P value |
|---|---|---|---|---|
| GP Patient Survey Scores* | Access | Random effect (log OR) from logistic case-mix adjustment model | −0.96 (−1.21 to −0.70) | <0.001 |
| | Continuity of care | | −0.09 (−0.25 to 0.07) | 0.274 |
| | Overall satisfaction | | −0.48 (−0.70 to −0.27) | <0.001 |
| Baseline workforce† | Ratio of nurse FTE to GP FTE | | 1.02 (−0.05 to 2.09) | 0.062 |
| | Adjusted weighted list size per GP FTE | Per 1000 patients per GP FTE | 0.40 (0.18 to 0.62) | <0.001 |
| | Total GP FTE | | −0.17 (−0.25 to −0.10) | <0.001 |
| | Ratio of 'Other' GP FTE to total GP FTE | | 0.65 (0.32 to 0.98) | <0.001 |
| Rurality setting‡ | Urban practice | | Reference | 0.404 |
| | Rural practice | | −0.13 (−0.43 to 0.17) | |
| Index of Multiple Deprivation—practice in quintile‡ | 1—least deprived | | Reference | <0.001 |
| | 2 | | 0.02 (−0.29 to 0.32) | |
| | 3 | | 0.13 (−0.16 to 0.42) | |
| | 4 | | 0.57 (0.29 to 0.85) | |
| | 5—most deprived | | 0.36 (0.06 to 0.66) | |
| Projected quantities | Adjusted weighted list size§ | Per 1000 patients | 0.14 (0.10 to 0.18) | <0.001 |
| | Proportion of GP FTE still in patient care* | Varies from 0 to 1 | 0.38 (−0.78 to 1.54) | 0.520 |
| | Proportion of GP FTE still in patient care × Ratio of nurse FTE to GP FTE* | | −1.01 (−2.48 to 0.46) | 0.177 |
| Constant | | | −4.15 (−5.10 to -3.21) | <0.001 |

*Data from 2012.
†Data from 2012 except nurse data which were from 2013.
‡Index of Multiple Deprivation data from 2016 for variable where this status is expected to remain relatively constant over time.
§Actual list size from 2016 rather than projected list size based on the 2012 data as pre-2012 data did not allow projections comparable to those which were made with more current data looking forward.
FTE, full-time equivalent; GP, general practitioner.

## DISCUSSION
### Summary of main findings
We developed an approach to modelling an individual general practice's future risk of being in a state of GP workforce undersupply. Within that work, we developed a 'main' model and a 'simpler' model. The 'main' model produced a range of risk scores attributable to practices across South West England, and, based on the ROC curve analysis, had a fair to good discriminatory ability. Applying our modelling approach suggests that the practices at highest risk of future undersupply of GP workforce are those which currently have, on average: larger patient lists, employ more nurses relative to doctors, serve more deprived and younger populations or have considerably worse patient experience ratings when compared with national averages.

In an extension of our research, we also modelled scenarios where the recruitment of GPs was more difficult than at present and/or where practice populations increase dramatically beyond what would be expected from historical local trends (eg, through a new housing development). These scenarios did identify practices where risk profiles changed, sometimes substantially, but in general, it was the same practices in all scenarios that were at highest risk of future undersupply of GP workforce. This almost certainly reflects the fact that those most likely to have problems in the future are those which are currently experiencing difficulties. This was evident from the relatively good predictions from a simple model including only contributing variables (ie, workload per FTE GP and GPPS patient access scores); this model had an area under the ROC curve that was not substantially less than that of the 'main' model, which drew on a wider range of variables, some of which were not routinely available in published data. In particular, we found that inclusion of findings from our own survey of GPs' career intentions had very little impact on the predictions when compared with using expected future workforce projections based only on routinely available data regarding GPs' gender and age.

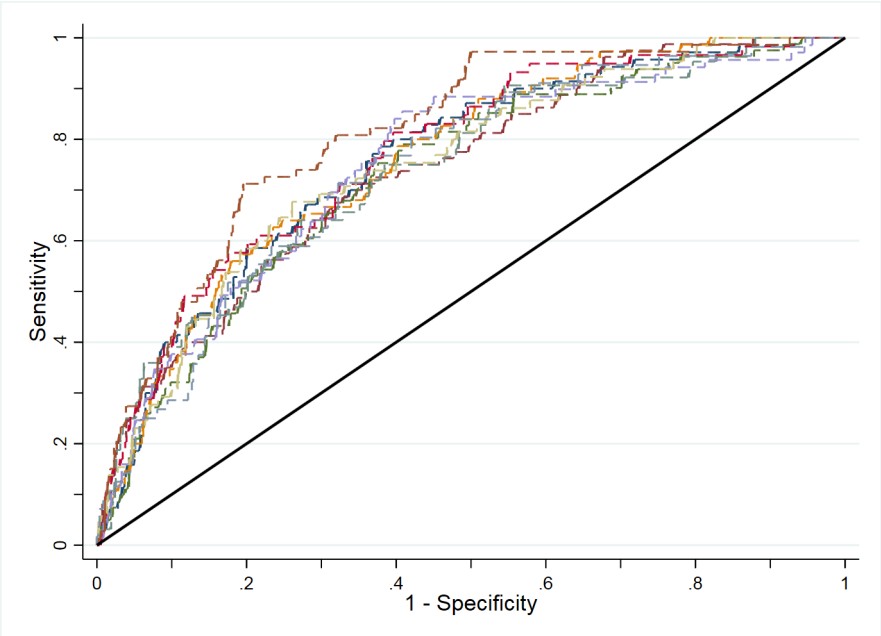

**Figure 2** Tenfold cross-validation receiver operating characteristic curve for the predictive risk model based on the national historical data used to build the model.

## Strengths and weaknesses

Strengths of this work include the comprehensive use of freely available data as well as the exploratory use of a census survey of career intentions of GPs in the region. The main strength is the novel development of factor weightings based on routinely available historical data. However, we recognise that this assumes that factors driving changes are constant from the historical time period of model development to the future time period of prediction. This is unlikely to be the case given recent problems in GP workforce recruitment and retention in the UK.[4] To this end, we have modelled what might be expected if recruitment was harder than it has been historically, and if there were substantive increases in the practice population. These scenarios may be more reflective of what we might expect going forward.

The main weakness of this work concerns our ability to distinguish in what situations, and in which practices, future GP workforce leaving patient care will impact the level of continuing GP workforce and its ability to meet

**Table 3** Differences between practices identified at high risk of future undersupply and other practices assuming a baseline scenario

| | High risk (n=92) | | | Other (n=276) | | | |
|---|---|---|---|---|---|---|---|
| | **Median** | **25%** | **75%** | **Median** | **25%** | **75%** | **P value*** |
| List size | 10 625 | 7732 | 13 195 | 6915 | 4941 | 10 206 | <0.001 |
| Adjusted weighted list size | 11 133 | 7369 | 13 252 | 7398 | 5251 | 10 615 | <0.001 |
| GP FTE | 5 | 3.1 | 6.6 | 4.5 | 3.1 | 6.6 | 0.445 |
| Ratio of nurse FTE to GP FTE | 0.7 | 0.5 | 1 | 0.4 | 0.4 | 0.6 | <0.001 |
| IMD | 25.6 | 18.7 | 31.7 | 17.6 | 13.1 | 22.2 | <0.001 |
| GPPS access† | 0.4 | 0.2 | 0.6 | 0.8 | 0.6 | 0.9 | <0.001 |
| GPPS continuity† | 0.3 | 0.2 | 0.5 | 0.7 | 0.5 | 0.9 | <0.001 |
| GPPS satisfaction† | 0.4 | 0.2 | 0.6 | 0.7 | 0.5 | 0.9 | <0.001 |
| % over 65 | 18.3 | 14.1 | 23.4 | 23.2 | 18.5 | 26.5 | <0.001 |
| Setting | **n** | **%** | | **n** | **%** | | **P value‡** |
| Urban practices | 77 | 31.3 | | 169 | 68.7 | | <0.001 |
| Rural practices | 15 | 12.3 | | 107 | 87.7 | | |

*From Mann-Whitney test.
†GPPS scores used were case-mix adjusted log ORs relative to the average practice nationally.
‡from Fisher's exact test.
FTE, full-time equivalent; GP, general practitioner; GPPS, General Practice Patient Survey; IMD, Index of Multiple Deprivation.

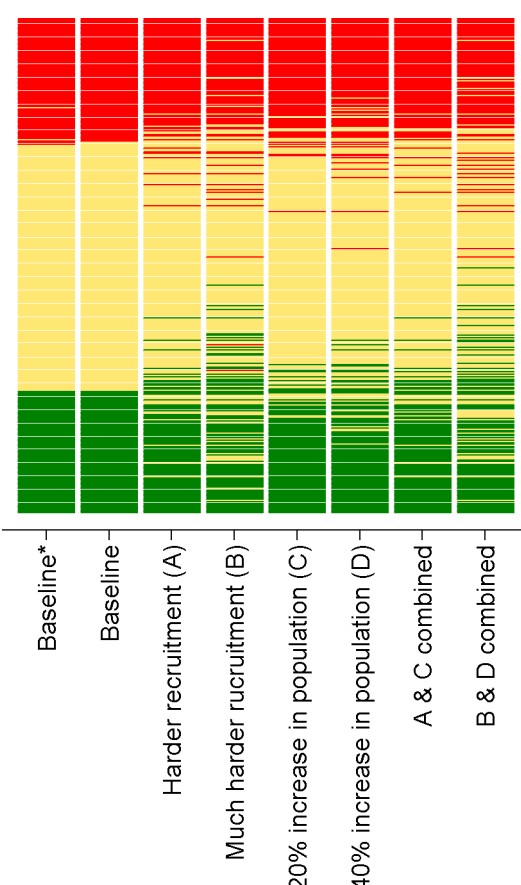

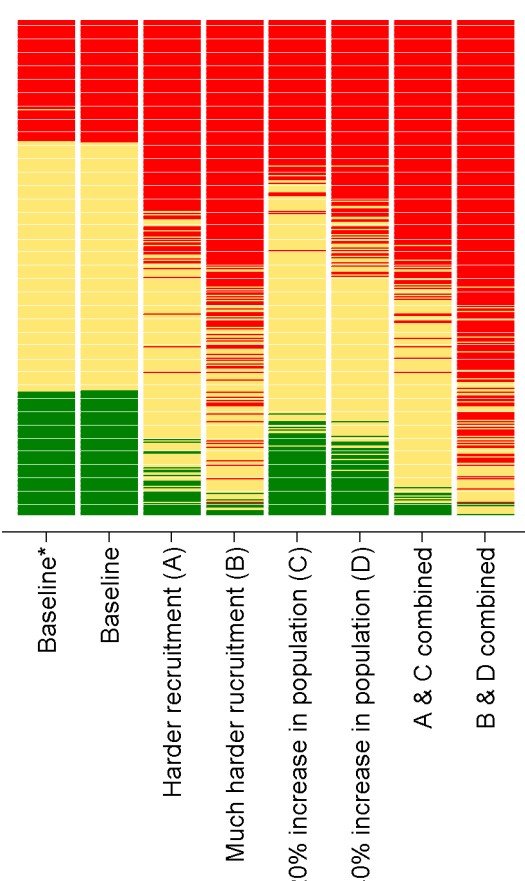

**Figure 3** Rating of practices in South West England from different risk prediction scenarios A–D using cut-offs defined by the quartiles of each prediction (relative risk). *Risk prediction as for baseline, but using age and gender of general practitioners alone rather than including responses to the career intentions survey. In each case, the practices are ordered by the baseline scenario.

**Figure 4** Rating of practices in South West England from different risk prediction scenarios A–D using cut-offs defined by the quartiles of the baseline prediction (absolute risk). *Risk prediction as for baseline, but using age and gender of general practitioners alone rather than including responses to the career intentions survey. In each case, the practices are ordered by the baseline scenario.

patient demand. For practices that do not encounter problems in recruiting GPs, retiring GPs pose much less of an issue than for practices where recruitment is difficult. Here we relied on the level of nurse staffing in a practice as a proxy for recruitment issues; importantly, this means the association of more nurses with at-risk practice status is likely to be attributable to practices being unable to fill GP vacancies, not that more nurses per se puts a practice at risk. A more direct measure of recruitment problems which was consistently and widely collected (such as duration of advertising for vacant posts, using a consistent methodology to track this) would be expected to provide a better model. Unfortunately, no robust freely available measure exists. The NHS GP census does collect data on time to fill vacancies[18] and existing unfilled vacancies. However, these data are not freely available, and, furthermore, are not mandatory for completion by practices.

Another weakness was that historical workforce data were not available in the same detail as current data (including nurse data not being available for 2012 at

all). This meant that future workforce predictions using historical data would not be as accurate as those using current data. These inaccuracies would lead to a loss of power, and potentially an attenuation of the associated regression coefficients. This may explain the low statistical significance of associated coefficients in the model.

Finally, we note that our assessment of the performance of our model was made on the same data the model was developed on, and thus may not be a reflection of the accuracy of future risk predictions. Validation of the future risk predictions would be welcome, but can only be undertaken in 5 years' time.

### Implications

We have demonstrated that it is possible to make reasonable predictions of an individual general practice's future risk of undersupply of GP workforce with respect to its patient population. With ongoing GP workforce issues in the UK, local models are being developed to identify potentially 'at-risk' practices.[26] However, unlike the

model we present here, it is not clear to what extent these models are evidence-based or to what extent their limitations are recognised by the users of the models or even what is meant by 'at risk'.

While the model we present here is set in the context of UK primary care, the general approach could be applied to other settings and in other locations. In all cases, the predictions will be inherently limited by the quality and quantity of available data. Improvements in data quality going forward will help the situation in the UK, particularly if data are released on GP recruitment. However, it will be some time before robust historical data exist that can be used for the model development process outlined here. If models such as the one outlined here are to be produced and used, it is important that high-quality data continue to be collected. However, it is worth recognising that the full range of data employed in the 'main' model produced only modest improvement in model fit over our 'simpler' model, suggesting that reasonable predictions may be made using a smaller number of variables. We have not attempted to establish a minimum useful set of data to make predictions of risk of undersupply of GP workforce. Rather, we have focused on an approach by which such predictions can be made. Given that, the lack of availability of variables such as those used here should not present a barrier to developing a model along similar lines suitable for other settings.

The predictions produced by this model and similar models may facilitate targeting of interventions to retain and attract GP workforce either in specific practices, or in specific regions currently at high risk of problems driven by workforce supply. Although our model provides reasonable discrimination, much could potentially be achieved by focusing efforts on those practices currently experiencing difficulties.

While a policy of targeted interventions may have value, we find that most practices are likely to be at a high risk of workforce undersupply when faced with a substantial increase in demand from an increased patient population combined with major difficulties in recruiting GPs. As such, local knowledge of drivers of increased practice populations, such as housing developments, will be key to being able to suitably apply targeted interventions. Even in South West England where workload and the ability to meet patient demand are better than in England overall, most practices are currently vulnerable to recruitment challenges, and will remain so going forward. Given this, national or broad regional policies and strategies may be more effective than targeted ones, especially if there is limited knowledge on how local populations are likely to evolve.

**Contributors** GAA, JLC, AS and NM conceived the study. GAA, MGC and NM performed analysis. GAA, MGC, NM, AS, EF, CS, RC, SGD, SHR, FW and JLC contributed to the design and interpretation of the study. GAA drafted the initial manuscript. GAA, MGC, NM, AS, EF, CS, RC, SGD, SHR, FW and JC commented and critically reviewed the manuscript prior to submission.

**Funding** The project was funded by the National Institute for Health Research, Health Service and Delivery Research programme (project 14/196/02). The views and opinions expressed herein are those of the authors and do not necessarily reflect those of the Health Service and Delivery Research programme, the National Institute for Health Research, the National Health Service or the Department of Health.

**Competing interests** All authors have completed the ICMJE uniform disclosure form at www.icmje.org/coi_disclosure.pdf (available on request from the corresponding author) and declare grants from the National Institute for Health Research (NIHR) (Health Service and Delivery Research programme) during the conduct of the study. SGD's position is partly supported by the NIHR Collaboration for Leadership in Applied Health Research and Care South West Peninsula at the Royal Devon and Exeter NHS Foundation Trust. The views expressed are those of the author(s) and not necessarily those of the NHS, the NIHR or the Department of Health.

**Patient consent for publication** Not required.

**Ethics approval** Ethics approval for the GP Census survey was provided by the University of Exeter Medical School Research Ethics Committee. All other data were publicly available and so ethical approval was not required for its use.

**Provenance and peer review** Not commissioned; externally peer reviewed.

**Data availability statement** Most data used in this study are publicly available from referencedsources. Data from the GP Census survey can be made available on request from the corresponding author of the original publication at john.campbell@exeter.ac.uk.

**Open access** This is an open access article distributed in accordance with the Creative Commons Attribution 4.0 Unported (CC BY 4.0) license, which permits others to copy, redistribute, remix, transform and build upon this work for any purpose, provided the original work is properly cited, a link to the licence is given, and indication of whether changes were made. See: https://creativecommons.org/licenses/by/4.0/.

**ORCID iDs**

Gary A Abel http://orcid.org/0000-0003-2231-5161
Emily Fletcher http://orcid.org/0000-0003-1319-3051
Chris Salisbury http://orcid.org/0000-0002-4378-3960
Sarah Gerard Dean http://orcid.org/0000-0002-3682-5149

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
