## [Reviewer comments · BMJ Open]

ARTICLE DETAILS

TITLE (PROVISIONAL)	Workforce predictive risk modelling – development of a model to identify general practices at risk of a supply-demand imbalance.
AUTHORS	Abel, Gary; Gomez-Cano, Mayam; Mustafee, Navonil; Smart, Andi; Fletcher, Emily; Salisbury, Chris; Chilvers, Rupa; Dean, Sarah; Richards, Suzanne; Warren, F; Campbell, John

VERSION 1 – REVIEW

REVIEWER	Rhema Vaithianathan Auckland University of Technology, New Zealand
REVIEW RETURNED	04-Feb-2019

GENERAL COMMENTS	Authors are predicting 2016 as the outcome year, I don't understand how they can use 2016 practice population as a predictor variable. The reason they have to do this is because they lack 2012 data - which suggests that what they propose (i.e. using a PRM for identifying practice with high imbalances) will not be readily applicable, Page 10 interpretation of negative co-efficient is not correct, as it is a MARGINAL effect holding everything else constant. This is a data mining exercise so it is not appropriate to interpret the coefficients. Statistical methods very poorly described, e.g: was the sample split and the accuracy measured on the hold-out sample?
--

REVIEWER	Lisa Smeds Alenius Karolinska Institutet, Sweden
REVIEW RETURNED	18-Feb-2019

GENERAL COMMENTS	I appreciate the opportunity to review this paper. The aim is interesting and the ambition to use freely available data to create workforce predictions is commendable as it would be valuable in improving workforce planning on both regional and national levels. In its current state, the paper give rise to several serious concerns, which are elaborated upon in the attached file. The main concern relates to the manner in which key measures are operationalised, which risks undermining the subsequent analysis and results. Summary This paper set out to develop a risk prediction model to identify general practices (GPs) at risk of experiencing shortages of doctors relative to the patient population's care needs. Using freely available data from several different sources and years, historical data is
---

applied on a current situation defined as baseline in order to predict risk of single GP-units, in South West England, experiencing shortages of doctors in the next five-year-period.

The aim is relevant and commendable as better workforce planning models on micro-levels are needed in order to improve predictions and specificity in national, regional and local health workforce planning enabling an appropriate delivery of healthcare services. However, due to limitations in available data and, in my view, inaccurate operationalisations of key measures, the subsequent analyses are based on questionable premises which in turn makes the results challenging to interpret and validate.

In its current state, the paper appear unfinished and needs considerable revising. It is unstructured in its presentation of key measures and methods of analysis. In addition, although offering supplemental material in an online appendix is a good way to make additional information available to interested readers, the authors makes excessive use of referring to supplemental material while leaving out key information in the paper. In my view, offering supplemental material does not exempt authors from putting necessary information in the paper, to allow the reader to understand, value and interpret study aim, design, analysis, methods, and results without the need to investigate any additional appendix.

To aid in my review and to reduce the risk of misunderstandings on my part, I have consulted with a statistician. Her main concern, as is mine, was the difficulty interpreting the text to decipher what the authors have done.

Major issues

1) **Operationalisation of key measures:** The authors' decision to exclusively use freely available data might be advantageous in terms of enabling analysis of data at a lower cost and perhaps also illustrating the potential use and value of such data. However, limitations in available data creates challenges in the need to define proxy measures which might threaten validity and reliability of the construct aimed to be analysed.

- a) Operationalisation of 'demand' – using the proxy measure of the *workload of doctors* which is calculated by using workload proxy measures, as follows:
 - i) Using *size of patient lists* at each GP rather than, which would be preferable, actual patient visits. Number of patients listed at a GP gives no information as to the proportion of patients actually using the GP services which would contribute to the workload of doctors.
 - ii) Using *access to services*, reported by

patients. While it might reflect degree of pressure on the general practice's ability to provide care services to patients, it might be challenging to identify the particular workload of doctors from the general workload of the general practice as a whole, presumably consisting of other healthcare professionals also providing care. One might argue that low access to care, might also reflect shortage of nurses and their care services in combination with the workload of doctors.

b) Operationalisation of 'supply' of doctors

- i) Using a survey to GP doctors. Information about the prior distribution of the survey is not sufficient in the current paper, sample size and year of survey needs to be added. In addition, in Appendix 1, description of the analysed items differ from that given in the paper, and the references are not correct throughout (e.g. order of numbered references compared to reference list, listing references not found in reference list).

Items on doctors' *future career intentions* were used for analysis. Although a bit unclear, items in the survey asked about doctors' intention to leave (workplace or career) within 2 or 5 years. In other research intention to leave has shown to be less reliable the further into the future the prediction is made, i.e. intention to leave within 2 years might have been better to capture actual intentions.

Authors also calculate doctors' intention to leave based on age and gender, although rationale for this is not apparent, is it perhaps as a measure of expected rate of retirement, where female doctors retire earlier than male doctors?

c) Measuring trouble recruiting doctors

- i) By using a ratio between number of full-time equivalent nurses to number of full-time equivalent doctors (rationale for this procedure needs to be given in the paper rather than in appendix). The authors' reasoning, which is somewhat unclear seems to be that a higher nurse:doctor ratio would reflect problems of recruiting doctors as more nurses might be hired to overtake some of the doctors' responsibilities. Although this might be accurate, a higher

nurse:doctor ratio might also be argued to reflect a potential organisational shift, where many countries are expanding the primary care sector and the role and responsibilities of nurses as well as doctors. A higher ratio might also reflect organizational decisions to hire more nurses to meet patient demand rather than difficulty recruiting doctors.

Data on nurse workforce appears to be imputed on data from 2012 – not providing information on e.g. turnover or changes in the area it is difficult to determine the appropriateness of this, whether it is reasonable or not to assume the data from 2013 can replace data from 2012.

ii) In terms of recruitment issues, it might have been more relevant to use average number of vacancies or turnover rates in the general practices to reflect recruitment challenges. Although I recognize such data might not be freely available.

2) Analysis and results

a) Due to the unclear presentation of variables and sources of data, it is challenging to follow along with the analysis and understand how the different variables were treated in the analysis, also because of discrepancies in what is said in different places in the text.

b) The term 'predictive risk modelling' is used in the paper and results are presented in terms of 'relative' and 'absolute' risk. However, I find it a bit misleading to use such terms when a traditional risk analysis does not appear to have been performed. Instead, the 'risk model' is based on a categorization of the outcome variable, the constructed binary variable 'under- supply', into high, medium and low risk of 'under-supply'. And stress-testing the model is done by changing different explanatory variables and investigating how the order of general practices change. In a traditional risk analysis, I would have expected to see the risk ratios presented in e.g. table 3, which is currently not the case.

i) In the results, the risk model refers to figures 2 and i) I suggest checking for accuracy in referring to the figures, e.g. p11, line 60. The figures lack information as to what the colours mean, although perhaps apparent if printed in colour, harder to see if printing in black/white, and also what the ranges are.

ii) The description of the 'stress testing scenarios', page 8 line 47 onward, is confusing. E.g. the sentence describing the first stress test – modelling an increase in the coefficient for expected proportion of doctors **remaining** in patient care, in parenthesis this is explained to imply a greater

	impact of doctors leaving patient care. c) The appropriateness of the way different data from different years have been applied to create the model is unclear. For instance (p7, under heading 'Development of predictive risk model'), using survey data from 2016 on doctors' career intention within 5 years to calculate expected proportion of doctors still in patient care in 2017, combined with data from patient survey (2011/12 and 2015/2016), nurse workforce (2013 and 2016), doctor workforce (2012 and 2016), patient lists (2014-2016), practice deprivation (2015) to create a baseline in 2016 of number of general practices in 'under-supply'. i) Regarding Appendix 2 - I recognize the challenge of illustrating the complex data flow, but I find the Appendix 2 to be confusing and not really helpful in understanding the process. Suggest omitting or revising. 3) Structure a) Suggest reconsidering the structure of presenting the methods as information seems repeated and sometimes slightly different in different sections, adding to the confusion. Consider the order of presenting/introducing information to the reader, making it succinct, avoid repeating information and ensure enough details are given in the text before referring to an appendix for additional information. b) I could not find any reference to ethical considerations Minor issues 4) Confusing use of abbreviations – GP appear sometimes to mean general practise, and sometime the general practitioner, i.e. the medical doctor. Suggest consistent use to avoid confusion. 5) Use of supplement materials - essential details missing in the text, such as the rationale for the decision to use total nurse FTE:total doctor FTE as a proxy for difficulty in recruiting doctors 6) Missing statement of Patient and Public Involvement in the method section
--	---

REVIEWER	Steffen Bayer University of Southampton, UK
REVIEW RETURNED	15-May-2019

GENERAL COMMENTS	This is a well written article introducing a model to predict GP practices likely to suffer undersupply over a five year time horizon. This is overall an interesting and worthwhile article reporting on careful work. There are however some areas which might deserve additional attention:
--

	 • The authors measure work-force imbalance if a practice is in the lowest third in access scores and the highest third nationally of list size per full-time equivalent GP. These are, of course, relative measures, not absolute measures and I would therefore have been interested in a discussion of advantages and disadvantages of such a relative measure given that it is likely that overall the supply situation is likely to become worse over the next 5 years. I would also have liked to understand how many practices would have been identified if only one of the two components of this measure had been used. • The authors use actual and not projected data (based on what was available in 2012) for practice populations in 2016 in their predictive model. They acknowledge the limitations and state the reasons for this approach. Nevertheless, it would have been interesting to see the performance a model entirely based on data available in the past (maybe using practice populations and a past growth rate even if not on practice level). A discussion of any differences would be useful given that it is not obvious what the influence of uncertainty on the demand side is. • The authors show that using a survey of career intentions or using age and gender specific estimates of quitting intentions does not make a big difference for the prediction of the risk of a demand-supply balance. This raises the question what the difference between predicted quitting probability with the two methods is (before using it as an input for the predictive model). However, given that the survey was not available for the 2012 data I wonder whether it would make more sense to remove the survey entirely from this paper. • The authors note that a weakness of their study was that the same data was used to develop and test their model. I would suggest considering using only a random sample of the practices to build the model and the use the remaining data to test the model. • The author mention that local models are being built to identify at risk models – it might be instructive to compare insights from these local model with insight form their model. • The data flow figure states at the bottom of the figure that 2012 data was used. This seems to be a typo – I assume it should be 2016.
--	--

REVIEWER	Sarah R Haile Epidemiology, Biostatistics and Prevention Institute, University of Zurich, Switzerland
REVIEW RETURNED	20-May-2019

GENERAL COMMENTS	This was a complex analysis of data from several combined sources of data to answer to relevant question about undersupply of general practitioners. The primary outcome was a combination of being both a) in the bottom 1/3 of access according to the GPPS and b) being in the highest 1/3 of workforce to workload ratio. This seems to be not specifically undersupply, but a combination of both limited access and undersupply. Are these two measures correlated in any way? It
---

	also seems a bit strange in the sense that with the 1/3 thresholds, by definition 1/9 of the practices will be defined as having undersupply. Is this really the case. Have you considered an analysis of just workforce to workload ratio (as a continuous measure)? That kind of analysis would permit you to quantify the amount of undersupply, rather than specifying that the top 1/3 of wf:wl ratio must be undersupply / overworked GPs, for which thresholds other than 1/3 could also be considered reasonable. Also, it may be interesting to consider a multivariate analysis with both the GPPS access questions and the wf:wl ratio as outcomes. In a minor related point, which GPPS questions were evaluated here? Regarding the data flow diagram: It seems that there is no clear distinction between raw data, intermediate variables, and the outcome in determining what is in the models. For example, access is both in the definition of undersupply and seems to be a predictor in the development model. And workforce:workload ratio is used in the definition of undersupply and is a predictor in the development model. Wouldn't this affect the interpretability of the model? It is also strange that the predictors in the development model are not the same the ones in the prediction model. Can you clarify if this was the case, and why? Also, in the usual training and testing setup for these kinds of validations, you generally have to leave some data out to check the predictions. Did you perform any such analysis? The flow chart of the data analysis was very interesting, though I had a very difficult time trying to match the methods description up with the diagram. The three colors for the arrows don't seem to help the readability here (though it is worse printed in black and white), and with the large number of arrows, it seems a bit like playing some untangling game. Is there any way to make this clearer to read? What was the "simpler model" you referred to? Could you show this in the flow chart? Could you have reduced the full model and still kept reasonable predictive properties? Area under the curve only measures discrimination, not calibration. Did you consider other performance measures? Data preparation > workload: It would be interesting to see the distribution of workload scores. Removing the top and bottom 2.5% seems overly conservative, even if some values couldn't possibly be correct. Do the results change much if you don't remove any, or only remove the top/bottom 1%? Were the "outliers" clustered in any sense (geographically? similar kinds of areas?, etc)
--	--

VERSION 1 – AUTHOR RESPONSE

Reviewer: 1	
Reviewer Name: Rhema Vaithianathan	
Page 10 interpretation of negative co-efficient is not correct, as it is a MARGINAL effect holding everything else constant. This is a data mining	Thank you for this comment. We have re-written the corresponding sentence to now read

exercise so it is not appropriate to interpret the coefficients.	“A negative coefficient implies a reduced risk of future undersupply as the value of the variable increases when all other variables are kept constant.” Whilst we fully agree with the author that the coefficients cannot be interpreted in a causal way this statement is still factually correct and it is important that the reader understands the direction of associations. For this reason we have decided to keep the statement in (having been modified).
Statistical methods very poorly described, e.g: was the sample split and the accuracy measured on the hold-out sample?	We have made numerous changes to the methods in response to the specific comments made by other reviewers. In response to this specific point we did note in the methods that the model development was made on national data (i.e. it was not split), but we have now rewritten the sentence to make this more explicit as follows “Model development was based on all available national data in order to maximise statistical power .We did not split the data into development and validation samples as changes over time in healthcare delivery are more likely to be a treat to future use of the model than over-fitting.” We also noted that this is a weakness in the discussion.
Reviewer: 2	
Reviewer Name: Lisa Smeds Alenius	
In its current state, the paper give rise to several serious concerns, which are elaborated upon in the attached file. The main concern relates to the manner in which key measures are operationalised, which risks undermining the subsequent analysis and results.	
Summary	

This paper set out to develop a risk prediction model to identify general practices (GPs) at risk of experiencing shortages of doctors relative to the patient population's care needs. Using freely available data from several different sources and years, historical data is applied on a current situation defined as baseline in order to predict risk of single GP-units, in South West England, experiencing shortages of doctors in the next five-year-period. The aim is relevant and commendable as better workforce planning models on micro-levels are needed in order to improve predictions and specificity in national, regional and local health workforce planning enabling an appropriate delivery of healthcare services. However, due to limitations in available data and, in my view, inaccurate operationalisations of key measures, the subsequent analyses are based on questionable premises which in turn makes the results challenging to interpret and validate.	We thank the reviewer for recognising the aim of the work is worthy of research. Whilst we do recognise that there are limitations of the data that we have used, we have aimed to be open and transparent about what we have done and the limitations of the approach and believe that the model produced does have value.
In its current state, the paper appear unfinished and needs considerable revising. It is unstructured in its presentation of key measures and methods of analysis. In addition, although offering supplemental material in an online appendix is a good way to make additional information available to interested readers, the authors makes excessive use of referring to supplemental material while leaving out key information in the paper. In my view, offering supplemental material does not exempt authors from putting necessary information in the paper, to allow the reader to understand, value and interpret study aim, design, analysis, methods, and results without the need to investigate any additional appendix. To aid in my review and to reduce the risk of misunderstandings on my part, I have consulted with a statistician. Her main concern, as is mine, was the difficulty interpreting the text to decipher what the authors have done.	We have made a number of changes to the paper which we hope have improved the presentation of the work. See below for specific responses/changes.
Major issues	

1) Operationalisation of key measures: The authors' decision to exclusively use freely available data might be advantageous in terms of enabling analysis of data at a lower cost and perhaps also illustrating the potential use and value of such data. However, limitations in available data creates challenges in the need to define proxy measures which might threaten validity and reliability of the construct aimed to be analysed.	Whilst we appreciate that there are limitations in the use of freely available data and that direct measures of all constructs of interest are not available, we feel that it is still of interest to demonstrate what can be done with these data. We do consider the use of a survey of GP quitting intentions as a direct measure, however, we find that it does not materially improve the fit of the model. We note that reviewer 3 has suggested removing this survey data from the model. We are not intending to do so, but contrasting that suggestion with this statement does illustrate the different points of view that exist in regards to the focus on routinely available data.
a) Operationalisation of 'demand' – using the proxy measure of the workload of doctors which is calculated by using workload proxy measures, as follows: i) Using size of patient lists at each GP rather than, which would be preferable, actual patient visits. Number of patients listed at a GP gives no information as to the proportion of patients actually using the GP services which would contribute to the workload of doctors. ii) Using access to services, reported by patients. While it might reflect degree of pressure on the general practice's ability to provide care services to patients, it might be challenging to identify the particular workload of doctors from the general workload of the general practice as a whole, presumably consisting of other healthcare professionals also providing care. One might argue that low access to care, might also reflect shortage of nurses and their care services in combination with the workload of doctors.	In developing our models we thought long and hard about how to conceptualise workload. Whilst it is true that patient visits might capture the actual work performed by a practice, the amount of work performed is inherently limited by the staff available to do the work. Thus there may be unmet demand within a practice. Thus by basing our measure on the registered population we obtain a measure of how much work a practice would be expected to do, given that population. We recognise that this logic was not made clear in the paper and we have added the following text to the workload section of the methods "We use a workload definition based on registered patients rather than recorded patient visits. Patient visits are a measure of actual work performed and to some degree are limited by the workforce available and as such cannot capture unmet demand. By considering the registered population we estimate the expected workload to serve that population based on national averages." In relation to the comment about a patient survey measure of access reflecting both issues with access to GPs and Nurses, we do accept that this may be true. However, it is important to recognise two things. Firstly, the majority of appointments within general practice are with GPs rather than nurse and secondly, that different practices employ nurses in different roles. It is for this reason that we have included the ratio of nurse FTE to GP FTE in our model. As such we consider it is right to consider an overall measure of the ability to provide care rather than specifically looking at access to GP appointments, as one way in which access to GPs might be improved is by employing nurses in more diverse roles.

b) Operationalisation of 'supply' of doctors i) Using a survey to GP doctors. Information about the prior distribution of the survey is not sufficient in the current paper, sample size and year of survey needs to be added. In addition, in Appendix 1, description of the analysed items differ from that given in the paper, and the references are not correct throughout (e.g. order of numbered references compared to reference list, listing references not found in reference list). (1) Items on doctors' future career intentions were used for analysis. Although a bit unclear, items in the survey asked about doctors' intention to leave (workplace or career) within 2 or 5 years. In other research intention to leave has shown to be less reliable the further into the future the prediction is made, i.e. intention to leave within 2 years might have been better to capture actual intentions. (2) Authors also calculate doctors' intention to leave based on age and gender, although rationale for this is not apparent, is it perhaps as a measure of expected rate of retirement, where female doctors retire earlier than male doctors?	The requested information has now been added to the main paper and appendix which now read as follows "Briefly, a questionnaire was administered to all active GPs in South West England in April-June 2016, enquiring about their intentions to cease/interrupt practice within 2 and 5 years (3370 questionnaires sent, 2248 returned, response rate 67%)." We have noted your point that the description of the considered survey items in the appendix differs from the main paper. The appendix does give more detail on the items, dealing with the first two questions separately from the third. However, having closely reviewed this the only inconsistency we can find is that the appendix does not note the free text response to the third item. This has now been edited. On the issue of whether we should use 2 and/or 5 year reported quitting intentions, we fully accept your point that the 2 year intentions may be more reliable than the 5 year ones. However, we are interested in the remaining workforce in 5-years' time. The 2-year quitting intentions may be a more reliable measure of actual quitting in 2-years, but are less likely to be a reliable measure of actual quitting in 5-years. Given our findings that age is the dominant drive of quitting intentions it is likely that GPs of a certain age may well intend to be working in 2 years' time but not in 5 years' time. For this reason we have not changed our approach. On the issue of the rationale for basing quitting intentions on age and gender - we have added some further details to the main paper for our overall rationale, though we do not feel that it requires a full justification given the space required. We do not explicitly expect different retirement ages between genders, but we do allow for the possibility. Where we are possibly more likely to see differences is quitting patient care below retirement age.
---	---

c) Measuring trouble recruiting doctors i) By using a ratio between number of full-time equivalent nurses to number of full-time equivalent doctors (rationale for this procedure needs to be given in the paper rather than in appendix). The authors' reasoning, which is somewhat unclear seems to be that a higher nurse:doctor ratio would reflect problems of recruiting doctors as more nurses might be hired to overtake some of the doctors' responsibilities. Although this might be accurate, a higher nurse:doctor ratio might also be argued to reflect a potential organisational shift, where many countries are expanding the primary care sector and the role and responsibilities of nurses as well as doctors. A higher ratio might also reflect organizational decisions to hire more nurses to meet patient demand rather than difficulty recruiting doctors. (1) Data on nurse workforce appears to be imputed on data from 2012 – not providing information on e.g. turnover or changes in the area it is difficult to determine the appropriateness of this, whether it is reasonable or not to assume the data from 2013 can replace data from 2012. ii) In terms of recruitment issues, it might have been more relevant to use average number of vacancies or turnover rates in the general practices to reflect recruitment challenges. Although I recognize such data might not be freely available.	We have moved the material discussing the rationale for including the interaction between the expected proportion of the GP workforce remaining in patient care and the nurse:GP FTE ratio into the main paper. And whilst we fully take your point that a more direct measure would be much more preferable, the data is simply not available. This limitation is made clear both in the methods (with the justification having been moved into the main paper) and in the discussion. As for the use of 2013 nurse data instead of 2012 data. We fully accept that this is an issue, but the 2012 data do not exist. We now explicitly mention this in the discussion when discussing the broader limitations of the poorer fidelity of the historic workforce data.
2) Analysis and results a) Due to the unclear presentation of variables and sources of data, it is challenging to follow along with the analysis and understand how the different variables were treated in the analysis, also because of discrepancies in what is said in different places in the text.	We have found some errors in both the main text and appendix 2 in terms of the data used in the model and have made amendments accordingly.

b) The term 'predictive risk modelling' is used in the paper and results are presented in terms of 'relative' and 'absolute' risk. However, I find it a bit misleading to use such terms when a traditional risk analysis does not appear to have been performed. Instead, the 'risk model' is based on a categorization of the outcome variable, the constructed binary variable 'undersupply', into high, medium and low risk of 'under-supply'. And stress-testing the model is done by changing different explanatory variables and investigating how the order of general practices change. In a traditional risk analysis, I would have expected to see the risk ratios presented in e.g. table 3, which is currently not the case.

i) In the results, the risk model refers to figures 2 and 3, I suggest checking for accuracy in referring to the figures, e.g. p11, line 60. The figures lack information as to what the colours mean, although perhaps apparent if printed in colour, harder to see if printing in black/white, and also what the ranges are.

ii) The description of the 'stress testing scenarios', page 8 line 47 onward, is confusing. E.g. the sentence describing the first stress test – modelling an increase in the coefficient for expected proportion of doctors remaining in patient care, in parenthesis this is explained to imply a greater impact of doctors leaving patient care.

We are somewhat confused by the comment that "traditional risk analysis does not appear to have been performed". Risk prediction models are often based on logistic regressions as we have performed here (and is appropriate for a binary outcome). Table 2 shows the regression coefficients from the logistic regression. These are log-odds ratios (rather than risk ratios). This provides the information required to reconstruct the model. We have chosen not to present odds ratios as given the wide range of units involved it is hard to make comparisons between the different variables, and, as reviewer 1 points out, these coefficients should not be interpreted as causal as they are marginal coefficients. In contrast Table 3 presents a simple descriptive comparison based on predicted risk, rather than based on the outcome.

We have added the following text to the stress testing scenarios section of the results to explain Figures 2 and 3 better (now numbered Figures 3 and 4 due to new figure).

"In Figure 3 practices coloured red (high risk) are in the top 25% of practices in terms of risk of undersupply for any given scenario, practices coloured green (low risk) are in the bottom 25% for any given scenario, with the middle 50% of practices coloured yellow. In Figure 4 practices coloured red (high risk) have an absolute risk of future undersupply greater than 10% (corresponding to the minimum absolute risk of future undersupply of the top 25% of practices in the baseline scenario), practices coloured green (low risk) have an absolute risk less than 2.8% (corresponding to the maximum absolute risk of the bottom 25% of practices in the baseline scenario) and intermediate practices are coloured yellow."

We hope this improves the understanding of the figures. The statement on line 60 was a typo and should have referred to Figure 3 and not Figure 2. This has been corrected

c) The appropriateness of the way different data from different years have been applied to create the model is unclear. For instance (p7, under heading 'Development of predictive risk model'), using survey data from 2016 on doctors' career intention within 5 years to calculate expected proportion of doctors still in patient care in 2017, combined with data from patient survey (2011/12 and 2015/2016), nurse workforce (2013 and 2016), doctor workforce (2012 and 2016), patient lists (2014-2016), practice deprivation (2015) to create a baseline in 2016 of number of general practices in 'under-supply'. i) Regarding Appendix 2 - I recognize the challenge of illustrating the complex data flow, but I find the Appendix 2 to be confusing and not really helpful in understanding the process.	We have tried to be open about what data we have used and have commented on the limitations of data availability for time periods we would have wanted to examine. On one specific point we use data from 2012 to predict the proportion of GPs expected to still be in patient care in 2017 (not 2016 data) we hope the re-written sections of the methods now make this clearer. Following this comment and that of other reviewers we have redrawn the data flow chart which we hope makes things clearer.
Suggest omitting or revising.	
3) Structure a) Suggest reconsidering the structure of presenting the methods as information seems repeated and sometimes slightly different in different sections, adding to the confusion. Consider the order of presenting/introducing information to the reader, making it succinct, avoid repeating information and ensure enough details are given in the text before referring to an appendix for additional information.	We have considered this request carefully. However, we are struggling to see a better structure. This is a complicated analysis with many linking sections, The current structure is dictated by the fact that a discussion of the data sources logically comes first and data preparation needs to come before the modelling section and the model use has to come last.
b) I could not find any reference to ethical considerations	The following ethics statement has been added "Ethics approval. Ethics approval for the GP Census survey was provided by the University of Exeter Medical School Research Ethics Committee. All other data was publically available and so ethical approval was not required for its use."
Minor issues	
4) Confusing use of abbreviations – GP appear sometimes to mean general practise, and sometime the general practitioner, i.e. the medical doctor. Suggest consistent use to avoid confusion.	Thank you for picking this up. GP implies General Practitioner in all cases apart from when discussing the GP Patient Survey. We solve this issue by describing using the full "General Practice Patient Survey" instead.
5) Use of supplement materials - essential details missing in the text, such as the rationale for the decision to use total nurse FTE:total doctor FTE as a proxy for difficulty in recruiting doctors	We have moved the suggested detail into the main document as suggested (see comment above).

6) Missing statement of Patient and Public Involvement in the method section	A PPI statement has now been added.
Reviewer: 3 Reviewer Name: Steffen Bayer	
This is a well written article introducing a model to predict GP practices likely to suffer undersupply over a five year time horizon. This is overall an interesting and worthwhile article reporting on careful work. There are however some areas which might deserve additional attention:	We thank the reviewer for this overall positive assessment.
The authors measure work-force imbalance if a practice is in the lowest third in access scores and the highest third nationally of list size per full-time equivalent GP. These are, of course, relative measures, not absolute measures and I would therefore have been interested in a discussion of advantages and disadvantages of such a relative measure given that it is likely that overall the supply situation is likely to become worse over the next 5 years. I would also have liked to understand how many practices would have been identified if only one of the two components of this measure had been used.	The reviewer is right that the use of relative measures to define an outcome introduces some interesting issues. We have added the following text to the outcome definition section of the methods to explore this. “Having used relative measures and cut points which were defined pragmatically for the purposes of this study in our definition of undersupply, we do not propose absolute and objective measures about whether a practice is ‘failing’ to deliver care. Indeed if provision of care were good everywhere and the supply of workforce were not an issue, such an approach would be inappropriate. However, in the current climate in the UK, this represents a pragmatic approach in the absence of a direct measure.” And the following text to the risk prediction section “It should be noted that although the original outcome definition was a relative one, the model treated them as absolute. In other words predictions obtained from the model identify the risk of having a workload to workforce ratio in 2021 higher than two-thirds of practices did in 2016 and a GPPS access score in 2021 lower than two-thirds of practices did in 2016. In the context of a nationally worsening situation this would allow for considerably more practices to be in a state of undersupply.” In terms of what would have happened if we had only used one of the two components used to define our outcome measure, reviewer 2 makes a similar point and we would refer back to our reply to that comment. Whilst this may have been an interesting exercise, we approached our outcome definition from a conceptual point of view and as such we feel the suggested analyses are beyond the scope of this paper.
The authors use actual and not projected data (based on what was available in 2012) for practice populations in 2016 in their predictive model. They acknowledge the limitations and state the reasons for this	Whilst we share the reviewer’s thoughts that this would be a good thing to do, we do not think that with the data available in 2012 we could make any meaningful comparisons and so have not included this.

approach. Nevertheless, it would have been interesting to see the performance a model entirely based on data available in the past (maybe using practice populations and a past growth rate even if not on practice level). A discussion of any differences would be useful given that it is not obvious what the influence of uncertainty on the demand side is.	
The authors show that using a survey of career intentions or using age and gender specific estimates of quitting intentions does not make a big difference for the prediction of the risk of a demand-supply balance. This raises the question what the difference between predicted quitting probability with the two methods is (before using it as an input for the predictive model). However, given that the survey was not available for the 2012 data I wonder whether it would make more sense to remove the survey entirely from this paper.	Whilst we accept the reviewers concerns, exploring the feasibility of incorporating survey responses from GPs about quitting intentions was one of the key objectives of this work and so we are keen to retain this in the paper.
The authors note that a weakness of their study was that the same data was used to develop and test their model. I would suggest considering using only a random sample of the practices to build the model and the use the remaining data to test the model.	Reviewer 1 also made this point and we refer back to our response to that point.
The author mention that local models are being built to identify at risk models – it might be instructive to compare insights from these local model with insight form their model.	Whilst we are aware of local models being developed we do not have access to them so we are unable to do this even though we agree with the reviewer that it would be a useful exercise.
The data flow figure states at the bottom of the figure that 2012 data was used. This seems to be a typo – I assume it should be 2016.	We thank the reviewer for spotting this. In response to comments by other reviewers we have reworked the data flow figure.
Reviewer: 4 Reviewer Name: Sarah R Haile	
This was a complex analysis of data from several combined sources of data to answer to relevant question about undersupply of general practitioners. The primary outcome was a combination of being both a) in the bottom 1/3 of access according to the GPPS and b) being in the highest 1/3 of workforce to workload ratio. This seems to be not specifically undersupply, but a combination of both limited access and undersupply. Are these two measures correlated in any way? It also seems a bit strange in the sense that with the 1/3 thresholds, by definition 1/9 of the practices will be defined as having undersupply. Is this really the case.	We have added a further figure showing the distribution of practices nationally and in South-West England across the two variables which defined our outcome. As can be seen from that figure, there is some correlation between the two variables. This is also apparent by the fact that we do not see 13.5% of practices being in under-supply which is higher than the 1/9th you suggested. We do note that this takes the paper over the recommended limit on tables and figures. If the editorial team feel that this is not justified we can move it into an appendix.
Have you considered an analysis of just workforce to workload ratio (as a continuous measure)? That kind of analysis would permit you to quantify the amount of undersupply, rather than specifying that the top 1/3	We did indeed consider this from the outset of the work. However, from our experience of working with GPs it was clear to us, that workforce to workload ratio was not sufficient and that some practices are able effectively

of wf:wl ratio must be undersupply / overworked GPs, for which thresholds other than 1/3 could also be considered reasonable. Also, it may be interesting to consider a multivariate analysis with both the GPPS access questions and the wf:wl ratio as outcomes. In a minor related point, which GPPS questions were evaluated here?	serve a given population with a lower resource than other practices and it was important to take that into account. The utility of our model would be diminished if we identified such practices as having a problem. The GPPS item used here was the access question "Last time you wanted to see or speak to a GP or nurse from your GP surgery: Were you able to get an appointment to see or speak to someone?" We have now included the wording of the GPPS items in the methods to make this clearer. And reiterate this in the outcome definition section of the methods and the first paragraph of the results by adding (ability to make an appointment) when describing the GPPS access measure.
Regarding the data flow diagram: It seems that there is no clear distinction between raw data, intermediate variables, and the outcome in determining what is in the models. For example, access is both in the definition of undersupply and seems to be a predictor in the development model. And workforce:workload ratio is used in the definition of undersupply and is a predictor in the development model. Wouldn't this affect the interpretability of the model? It is also strange that the predictors in the development model are not the same the ones in the prediction model. Can you clarify if this was the case, and why? Also, in the usual training and testing setup for for these kinds of validations, you generally have to leave some data out to check the predictions. Did you perform any such analysis?	You are correct that both raw variables and derived variables are used in the prediction model. This simply depends on whether raw variables are available to act as measures for the construct of interest. You are also correct that access and workforce:workload ratio are used in the definition of undersupply as the outcome and are also used as exposure variables. The crucial information here is the date from which the data are derived. In the development model 2016 data is used to form the outcome variable, whereas 2012 data are used for the exposure variables. In relation to the observation that there were different variables used in the development model and the prediction model we have discovered that there was an error in the diagram. An arrow from the 2016 GPPS access data to the prediction model was omitted. Also there were two arrow from the 2012 expected proportion of GP workforce remaining in patient care. In relation to the issue of checking prediction on some excluded data. We did not do this. See response to reviewer 1 on the subject and text in methods and discussion.
The flow chart of the data analysis was very interesting, though I had a very difficult time trying to match the methods description up with the diagram. The three colors for the arrows don't seem to help the readability here (though it is worse printed in black and white), and with the large number of arrows, it seems a bit like playing some untangling game. Is there any way to make this clearer to read?	We are glad that you have found the flow chart interesting. We realise that it is hard to follow, but this is mainly due to the complex flow of data. In order to try and simplify the diagram and improve readability we have split the flow diagram into two with the development model represented on one page and the prediction model on another.
What was the "simpler model" you referred to? Could you show this in the flow chart?	Thank you for this suggestion – we have now also included an additional flow diagram for the simpler model in appendix 2.
Could you have reduced the full model and still kept reasonable predictive properties?	As noted in the discussion the simpler model investigated achieves an AUROC of 0.718 compared to 0.759 from the full model. This

	implies the additional variables make a modest but meaningful improvement to the predictive value of the model. We could in theory look at model with intermediate complexity between these two extremes, but we consider this beyond the scope of the study
Area under the curve only measures discrimination, not calibration. Did you consider other performance measures?	We thank the reviewer for this suggestion. We now include a calibration plot in a new appendix.
Data preparation > workload: It would be interesting to see the distribution of workload scores. Removing the top and bottom 2.5% seems overly conservative, even if some values couldn't possibly be correct. Do the results change much if you don't remove any, or only remove the top/bottom 1%? Were the "outliers" clustered in any sense (geographically? similar kinds of areas?, etc)	Following this comment we have now performed a sensitivity analysis removing only the top/bottom 1% as suggested. The model coefficients are broadly in line with the presented model with the exception of the coefficient for the expected proportion of GP workforce to remain in patient care which sees a 43% reduction. This is likely due to the measurement error at the extremes of the distribution when including the extra practices. This is also discussed in the methods section adding the following text “We also performed a sensitivity analysis to examine the impact of excluding the top and bottom 2.5% of practices in terms of workload per GP FTE. To do so we re-ran the logistic regression after excluding only the top and bottom 1% of practices in terms of workload per GP FTE.” And in the results with the following text “The sensitivity analysis excluding only the top and bottom 1% of practices in terms of workload per GP FTE produced broadly similar regression coefficients with the exception of the coefficient for the expected proportion of GP workforce to remain in patient care which was reduced by 43% (results not shown).”

VERSION 2 – REVIEW

REVIEWER	Steffen Bayer Southampton Business School, Southampton University, UK
REVIEW RETURNED	26-Aug-2019

GENERAL COMMENTS	I commend the authors for improving their paper. The explanations of the model are still somewhat confusing. I would encourage the authors to improve the clarity further. I can't follow the figures in appendix 2 – 2016 and 2012 seem to be interchanged in several instances. It would also have been useful if the main text would specifically refer to the different figures in appendix 2 and not just globally to appendix 2 at the beginning of the data section. I don't understand why the "expected proportion of GP FTE still in patient care in 2017" is one of the predictors for the predictive risk model, shouldn't it be 2016? This measure would deserve some
---

	further explanation in the text of the paper regarding how this prediction was made based on 2012 data and what uncertainties are inherent in this process. It seems to me still that not including your survey on carrier intentions in this paper would be preferable since including this does not seem to improve your prediction or the clarity of the presentation. However, if you want to include the survey, I would also like to see a short discussion on the comparison of the actual workforce predictions itself using different methods especially given that we can't know yet what the accurate numbers in the future will be.
--	---

REVIEWER	Sarah R Haile University of Zurich, EBPI Switzerland
REVIEW RETURNED	19-Aug-2019

GENERAL COMMENTS	In the 3rd paragraph of the introduction it is stated that the aim of the research was to develop a method to identify NHS practices which may face supply-demand workforce imbalances. As it stands currently, there are two main remaining issues with the analysis in terms of reaching this stated goal: 1) Because there were no testing or training data sets, the observed results are overfit to the 2012 data, and don't really allow for generalization to other datasets (other geographic regions, other time points, etc). This analysis presented here would be much stronger if some sort of data-splitting or cross-validation was used. 2) No discussion was made of the properties of the full vs simpler models. That is, would a simpler (or intermediate) model be sufficient to identify NHS practices with supply-demand workforce imbalances? This is an important question, especially if the evaluated model is to be used in other settings, especially where all variables may not be available. For example, rurality appeared to not be statistically significant in the model, nor did "proportion of GP FTE still in patient care", or the "continuity of care" variable. A simpler model would make your findings more generalizable to other settings, therefore having a greater impact. Thank you for the new Figures related to the data flow diagram, they are much clearer. On rereading this manuscript, I, like some of the other reviewers, found the methods sections to be quite long and hard to follow. While I see that you have a lot of important details in there, I'm wondering if there is not a way to make it easier to digest. To that end, I have a few suggestions:  - The first paragraph of the overview subsection is really more appropriate material for the introduction to the paper (especially as that is the only section prior to the methods). - The second paragraph of the overview could be substantially moved to the part about the model. - I would recommend to shorten the section "Data Sources" to simple descriptions of the data sources, without listing the questions
--

	that were derived from them. To aid in the description of the items later, each source should have a short name ("workforce") or acronym ("GPPS").  - In the section "Variables" ("data preparation and variable creation" is a bit long and confusing, and "data preparation" is understood without stating it), start with the outcome variable, and give the data source of the questions relating to it here. This would also be a good place to describe the GPPS questions used. For each of the other variables described, mention the source of that data and any specific questions used here. - The 3 proxy measures under "development of predictive risk model" should also be moved to the variables section. - First paragraph of "future risk prediction": This a very roundabout paragraph, making to difficult to see exactly what was done. Perhaps some of the reasons and other options described here should be moved to another section? For example, the discussion? (That is actually true of many of the "we also considered X, Y and Z" sections within the methods.) - Consider moving the "stress-testing" scenarios to another subsection called "sensitivity analysis". - Finally, what software did you use?
--	---

VERSION 2 – AUTHOR RESPONSE

Reviewer: 3	
Reviewer Name: Steffen Bayer	
Institution and Country: Southampton Business School, Southampton University, UK Please state any competing interests or state 'None declared': None	
I commend the authors for improving their paper.	We thank the reviewer for recognising the improvements, and for identifying areas where further clarification was needed. We hope the changes that we have made fulfil the reviewer's requirements.
The explanations of the model are still somewhat confusing. I would encourage the authors to improve the clarity further. I can't follow the figures in appendix 2 – 2016 and 2012 seem to be interchanged in several instances. It would also have been useful if the main text would specifically refer to the different figures in appendix 2 and not just globally to appendix 2 at the beginning of the data section.	There were some typographical errors in some date labels in Appendix 2; we thank the reviewer for spotting these. We have corrected these and hopefully as a consequence these figures should now be easier to follow. We have now also referred to the different figures in the appendix in the text as suggested. Furthermore, as you will see from our response to reviewer 4 we have made further changes to the text to aid clarity.
I don't understand why the "expected proportion of GP FTE still in patient care in 2017" is one of the predictors for the predictive risk model, shouldn't it be 2016? This measure would deserve some further explanation in the text of the paper regarding how this prediction was	As you will see from our answer to the point below it was always a key aim of this work to assess the utility of a survey on career intentions in predictions of undersupply status. Given our survey asks about quitting intentions over a 5 year timescale we felt it was important to reflect

made based on 2012 data and what uncertainties are inherent in this process.	this in all predictions of future workforce, such that the two methods (i.e. using publicly available data only, and using publicly available data plus the GP survey data) could be used interchangeably.
It seems to me still that not including your survey on carrier intentions in this paper would be preferable since including this does not seem to improve your prediction or the clarity of the presentation. However, if you want to include the survey, I would also like to see a short discussion on the comparison of the actual workforce predictions itself using different methods especially given that we can't know yet what the accurate numbers in the future will be.	It has always been a major aim of this work to assess the utility of a survey of career intentions on the ability to predict general practices risk of future workforce problems. On face value it would seem an invaluable addition. However, what we in fact show is that it adds very little value beyond knowing the age and gender of GPs at the practice. We feel strongly that the observation that the survey does not impact predictions is itself a very useful finding. For this reason we have chosen to keep these results but have taken on board your useful suggestion of introducing a short comparison of the two methods. To that end we have edited the relevant paragraph in the results to read “First, we examine the changes in predictions when using the two different methods of quantifying the likely future GP workforce remaining in patient care (one method using the results of the career intention survey and one method using only on GP age and gender). The two methods produced similar values for the likely proportion of GP workforce remaining in patient care with a Spearman correlation of 0.77 between the estimates made using the two methods in the 387 practices with at least one survey response. When using the different methods in the risk prediction model, there was very little difference in practices categorised as being either at “high relative risk” or “high absolute risk” of undersupply (seen in Figure 4 as limited reclassification of practices, correlation of ranks=0.999).”
Reviewer: 4	
Reviewer Name: Sarah R Haile	
Institution and Country: University of Zurich, EBPI, Switzerland Please state any competing interests or state ‘None declared’: None declared	
Thank you for your revision.	
In the 3rd paragraph of the introduction it is stated that the aim of the research was to develop a method to identify NHS practices which may face supply-demand workforce imbalances. As it stands currently, there are two main remaining issues with the analysis in terms of reaching this stated goal:	We would like to thank the reviewer for their comments with the aim of clarifying our paper. In particular we are grateful for the directive nature of many comments which are very clear in what changes are being suggested. We have considered all of these suggestions carefully and have made a number of changes. We do note that some pieces of information that the reviewer has suggested trimming back on were added at

	the request of previous reviewers. Although the statistical methods employed in this piece of work are not particularly complex, the data preparation and flow does involve many steps and there is a limit to how simplified it can be. Different reviewers and readers will have different opinions on the balance between clarity and the level of detailed provided and we trust that our latest effort meets a happy medium between the two.
1) Because there were no testing or training data sets, the observed results are overfit to the 2012 data, and don't really allow for generalization to other datasets (other geographic regions, other time points, etc). This analysis presented here would be much stronger if some sort of data-splitting or cross-validation was used.	We agree with the reviewer that employing a technique such as data splitting or cross validation would improve the generalisability of the findings. In order to make maximal use of the data available we use 10-fold cross validation when calculating the area under the ROC curve. In fact it makes very little difference changing from 0.759 to 0.755 for the full model and from 0.718 to 0.695 for the simpler model. We have made changes to the methods and results to reflect this (including figure 2). We have also changed the final paragraph of the strengths and limitations section of the discussion to read as follows “Finally, we note that our assessment of the performance of our model was made on the same data the model was developed on, and thus may not be a reflection of the accuracy of future risk predictions. Validation of the future risk predictions would be welcome, but can only be undertaken in 5 years’ time.”
2) No discussion was made of the properties of the full vs simpler models. That is, would a simpler (or intermediate) model be sufficient to identify NHS practices with supply-demand workforce imbalances? This is an important question, especially if the evaluated model is to be used in other settings, especially where all variables may not be available. For example, rurality appeared to not be statistically significant in the model, nor did "proportion of GP FTE still in patient care", or the "continuity of care" variable. A simpler model would make your findings more generalizable to other settings, therefore having a greater impact.	Whilst we appreciate the sentiment of the reviewer, we also recognise that the paper is already very complex. We felt strongly that introducing further intermediate models to the paper at this stage would only serve to increase the complexity of the paper further and so have not done so. However, we have taken on board the sentiment of the comment and we have made a number of edits to the paper accordingly. We have edited the first two paragraphs of the discussion to shift the focus to the general approach, rather than the specific model we have generated. The reviewer is right that it may not be possible to recreate a model with all data sources as we have to hand in other settings. In fact we feel that it is inevitable that it would not be possible, rather that the data available would be highly setting dependant. We feel that our paper demonstrates an approach which has utility and exactly what data sources are used will depend on both the setting and the data that is available. This will no doubt also change over time. To this end we have also added the following text to the second paragraph of the limitations section. “However, it is worth recognising that the full

	range of data employed in the 'main' model produced only modest improvement in model fit over our 'simpler' model, suggesting that reasonable predictions may be made based using smaller number of variables. We have not attempted to establish a minimum useful set of data to make predictions of risk of undersupply of GP workforce. Rather, we have focused on an approach by which such predictions can be made. Given that, the lack of availability of variables such as those used here should not present a barrier to developing a model along similar lines suitable for other settings."
Thank you for the new Figures related to the data flow diagram, they are much clearer.	We thank the reviewer for this comment
On rereading this manuscript, I, like some of the other reviewers, found the methods sections to be quite long and hard to follow. While I see that you have a lot of important details in there, I'm wondering if there is not a way to make it easier to digest. To that end, I have a few suggestions:	As we said above we very much appreciate the clear guidance provided by the reviewer. We give detailed responses to each suggestion below
- The first paragraph of the overview subsection is really more appropriate material for the introduction to the paper (especially as that is the only section prior to the methods).	We have moved this paragraph as suggested.
- The second paragraph of the overview could be substantially moved to the part about the model.	We thought hard about this suggestion. This paragraph was introduced in the drafting section at the suggestion of one of the co-authors who found the paper hard to follow when the methods launched straight into the data sources with no idea of why they were being used. We appreciate that different readers may have different preferences on this front, but we feel on reflection that the overview is useful in its current position.
- I would recommend to shorten the section "Data Sources" to simple descriptions of the data sources, without listing the questions that were derived from them. To aid in the description of the items later, each source should have a short name ("workforce") or acronym ("GPPS").	We have followed this suggestion and shortened sections for the GPPS, workforce GP quitting intentions and Subnational population projections data source descriptions, with much of the material moving to variables. We have however, decided not to use a short name or acronym as we feel this could lead to confusion rather than clarity.
- In the section "Variables" ("data preparation and variable creation" is a bit long and confusing, and "data preparation" is understood without stating it), start with the outcome variable, and give the data source of the questions relating to it here. This would also be a good place to describe the GPPS questions used. For each of the other variables described, mention the source of that data and any specific questions used here.	We have renamed the section Variables as suggested. As for starting with the outcome variable, whilst we appreciate the logic with which this suggestion was made, the outcome variable is itself essentially a composite of two other variables (although formed using data from a different time point). As such it is very hard to define the outcome variable first. We have given detail of specific questions used in this section as suggested
- The 3 proxy measures under "development of predictive risk model" should also be moved to the variables section.	The proxy measures referred to here are not in fact new variables, but existing ones which are already in our model. The exploratory analysis involved the inclusion of an interaction between a proxy measure and the expected future GP workforce. We appreciate this may not have

	been clear and have rewritten the corresponding paragraph as follows “We recognised the need to account for the fact that GPs leaving patient care would be most likely to impact the supply–demand balance when recruitment of staff was difficult. The implication of this is that ideally an interaction between future expected GP workforce and a measure of recruitment difficulty would be included in the model. However, we were unable to obtain any direct measure of the difficulty any one practice had in recruitment and so instead we explored the use of three variables already included as proxy measures for this: “
- First paragraph of "future risk prediction": This a very roundabout paragraph, making to difficult to see exactly what was done. Perhaps some of the reasons and other options described here should be moved to another section? For example, the discussion? (That is actually true of many of the "we also considered X, Y and Z" sections within the methods.)	We have simplified the relevant paragraph in response to this comment and it now reads “We made predictions of the future risk of undersupply of GP workforce for all practices in South West England with available data (being those included in the survey of future career intentions). The coefficients from the historical model were applied to 2016 data to form our baseline risk predictions with a 5-year forward view (as shown in Appendix 2b). Practices in the highest 25% of the predicted risk profile were flagged as “high risk” of future under-supply of GP workforce, those in the lowest 25% were flagged as being “low risk”, and those in between were flagged as being at “moderate risk”.”
- Consider moving the "stress-testing" scenarios to another subsection called "sensitivity analysis".	We did consider this suggestion but disagree with the reviewer that these are sensitivity analyses. Our understanding is that a sensitivity analysis explores what would happen to your findings if you had made a different assumption or had processed data in a different way. The scenarios in question are exploring the way in which predictions change when parameters of the model or expectations of future workload are changed. As such they are not what we understand to be sensitivity analyses.
- Finally, what software did you use?	The following sentence has been added to the end of the results section “Analyses were performed using Stata V14 and V16 and the 10-fold cross-validation was performed using the CVAUROC command.”

VERSION 3 – REVIEW

REVIEWER	Steffen Bayer Southampton University, UK
REVIEW RETURNED	01-Nov-2019

GENERAL COMMENTS	Thank you for these revisions. I am satisfied that you have fully addressed my concerns.
--

REVIEWER	Sarah R Haile Epidemiology, Biostatistics and Prevention Institute, Switzerland
REVIEW RETURNED	28-Oct-2019

GENERAL COMMENTS	Thank you for your very detailed revision.
--